# Impact of Chronic Exposure to Endometriosis on Perinatal Outcomes: Establishment of a Mouse Model

**DOI:** 10.3390/biomedicines10102627

**Published:** 2022-10-19

**Authors:** Mohammed Elsherbini, Kaori Koga, Eiko Maki, Keiichi Kumasawa, Erina Satake, Ayumi Taguchi, Tomoko Makabe, Arisa Takeuchi, Gentaro Izumi, Masashi Takamura, Miyuki Harada, Tetsuya Hirata, Yasushi Hirota, Osamu Wada-Hiraike, Yutaka Osuga

**Affiliations:** 1Department of Obstetrics and Gynecology, Faculty of Medicine, The University of Tokyo, Tokyo 113-8654, Japan; 2Department of Obstetrics and Gynecology, Faculty of Medicine, Saitama Medical University, Saitama 350-0495, Japan; 3Department of Integrated Women’s Health, St Luke’s International Hospital, Tokyo 104-8560, Japan

**Keywords:** pregnancy, fertility, mating, newborn, resorption, SGA, preterm labor, delivery, age, preconception

## Abstract

The purpose of this study was to establish a new mouse model of endometriosis that mimics real-world women’s health problems, in which women continue to be affected by endometriosis long before they wish to become pregnant, and to evaluate the impact of “chronic exposure to endometriosis” on perinatal outcome. Endometriosis was established by the intraperitoneal injection of homologous minced mouse uteri. Vehicle was injected for the control. Mating was initiated either 1 or 43 days after disease establishment (Young or Aged studies, respectively). Mice were sacrificed on 18 dpc. The number pups and resorptions were counted and pups’ body weights (BW) were measured, and the endometriosis lesion was identified and weighted. In the Young study, the number of resorptions and BW were comparable between the groups. In the Aged study, the number of resorptions was significantly higher and BW was significantly lower in endometriosis than that in control. The total weight of endometriosis lesion per dam was significantly lower in the Aged compared to the Young endometriosis group; however, not a single mouse was found to have any lesions at all. These results suggest that in addition to the presence of endometriosis per se, “chronic exposure to endometriosis” prior to pregnancy affect perinatal outcomes.

## 1. Introduction

Endometriosis affects approximately 10% of women of reproductive age and can cause severe chronic pelvic pain and infertility [1]. In addition to pain and infertility, endometriosis has recently been found to have a negative impact on perinatal outcomes, such as preterm labor, placental previa and being small for gestational age [2,3,4]. However, it is still not very clear how endometriosis causes such negative effects during pregnancy.

To date, various rodent models of endometriosis have been established to elucidate its pathophysiology [5,6,7,8] and to test the efficacy of the treatment [9,10,11]. On the other hand, only a few rodent models have been reported to evaluate endometriosis-related adverse perinatal outcomes [12,13,14,15,16]. In addition, all of these models were mated and conceived soon after establishing endometriosis lesions, and to date, no studies have examined the impact of long-term morbidity with endometriosis on pregnancy outcomes.

In recent years, the concept that “endometriosis is a chronic disease” has been proposed, and its lifelong management has been recommended [17]. In particular, the age of onset of endometriosis tends to be earlier [18,19,20], while the age at which a woman wishes to conceive tends to be later [21], meaning that the period of chronic exposure to endometriosis before women becomes pregnant is increasing, necessitating optimization of management for endometriosis in the preconception period. However, the impact of chronic exposure to endometriosis on perinatal outcomes and its mechanisms remain unclear, making the optimization difficult.

The purpose of this study was to establish a new mouse model of endometriosis that mimics real-world women’s health problems, in which women continue to be affected by endometriosis long before they wish to become pregnant, and to evaluate the impact of “chronic exposure to endometriosis” on perinatal outcome.

## 2. Materials and Methods

### 2.1. Establishment of Endometriosis Mouse Model

All procedures described in this study were conducted in accordance with the guidelines and regulations of the Animal Care and Use of the University of Tokyo Committee. Six-week-old BALB/c female mice were purchased from Japan SLC, Inc. (Tokyo, Japan). Mice were fed on a mouse diet and water and maintained on a light/dark cycle (12 h/12 h) under controlled living conditions. The endometriosis mouse model was established according to the published protocol [22,23]. For instance, uterine tissues were obtained from homologous donor mice, minced into small fragments under sterile conditions, and injected into the peritoneal cavity of the recipient mice (day 0) at a ratio of ½ uterus (1 horn)/recipient mouse, as shown in the diagram (Figure 1). Unlike previously published protocols, neither donor nor recipient mice received estradiol because this model is used for perinatal outcome analysis. In the control group, PBS was injected intraperitoneally. On the day of sacrifice, the endometriosis lesion was identified and weighed.

### 2.2. Establishment of Pregnancy Mouse Model

To determine the effect of the time between the onset of endometriosis and the desire to conceive on the perinatal outcome, two experimental groups were prepared with different time periods between the establishment of endometriosis and the initiation of mating (Figure 1). In the Young study, after establishing the endometriosis (day 0) mating was initiated on day 1 and continued every 4 days until day 42. In the Aged study, the mating was initiated on day 43 and continued every 4 days until day 84. To describe the experiment based on their week age, endometriosis was established or sham-operated at 7 weeks of age, and the mice were mated for 6 weeks each, from 7 to 12 weeks of age in the Young study and from 13 to 18 weeks of age in the Aged study. The ratio at mating was 1:2 (male:female) and male mice were swapped at each mating.

### 2.3. Fertility Analysis

As mentioned earlier, mating was initiated on day 1 or day 43 and repeated every 4 days until day 42 or day 84 (Young study, Aged study, respectively). Mice that did not conceive by the last mating were considered infertile. The infertile mice were sacrificed at day 60 or day 102 (Young study, Aged study, respectively). The time to conception was shown in a Kaplan–Meier survival graph.

### 2.4. Perinatal Outcome Analysis

After the date of the positive plug, the body weight was tracked daily, and dam was sacrificed at 900 AM on dpc 18. Spontaneous delivery before dpc 18 was defined as preterm labor. At the sacrifice, the number of living pups (litter size) and resorptions were counted. The number of implantation sites was calculated as the sum of the litter size and the number of resorptions. Pups were cleaned from the uterus, pictured, and weighted. All dams with litter size between 4 and 12 were included in the analysis for all studies. Dams with litter size ≤ 3 or ≥ 13 were excluded from body weight analysis but included in other analysis.

### 2.5. Statistical Analysis

Kaplan–Meier analysis was used for fertility analysis. Wilcoxon test was used for perinatal outcome analysis. The data analysis was conducted using JMP Pro 15 software (SAS Institute Inc., Cary, NC, USA). *p* < 0.05 was considered statistically significant.

## 3. Results

### 3.1. Effect of Endometriosis on Fertility

Firstly, the pregnancy incidence was assessed. The time to conception was significantly longer in Young E (*p* < 0.05) than in Young C, while there was no significant difference in the time between Aged C and Aged E (Figure 2).

### 3.2. Effect of Endometriosis on the Onset of Labor

In this study, all dams were planned to be sacrificed on dpc 18 for analyzing their pups. Under this plan, none of the control dams experienced spontaneous labor and delivery before dpc 18, while 2 out of 29 (6.9%) dams in the Young E group (on dpc 13 and dpc 17) and 1 of 39 (2.6%) dams in the Aged E group (on dpc 17) experienced spontaneous labor and delivered, respectively (Table 1).

### 3.3. Effect of Endometriosis on Neonatal Outcome

#### 3.3.1. Morphological Changes of Pups and Placenta

Macroscopically, there were no obvious abnormalities on the placenta and pups or growth restriction, but pups in the endometriosis group appeared to be slightly smaller and had more resorptions (Figure 3A,B), as evidenced by the statistical analysis described below.

#### 3.3.2. Litter Size

The litter size was not significantly different between the groups (7.2 ± 0.4, 7.3 ± 0.6, 7.3 ± 0.4 and 7.2 ± 0.4, mean ± SEM, Young C, Young E, Aged C and Aged E, respectively) (Figure 3C). The number of resorptions in each group was as follows (2.3 ± 0.3, 2.4 ± 0.3, 1.7 ± 0.2, 2.4 ± 0.2, mean ± SEM, Young C, Young E, Aged C, Aged E, respectively). The number of resorptions was significantly higher in the Aged E group compared to the Aged C group (*p* < 0.05) while there was no difference between the other groups (Figure 3D). The number of implantation sites did not differ significantly between the groups (9.5 ± 0.4, 9.7 ± 0.5, 9.0 ± 0.4, 9.7 ± 0.4, mean ± SEM, Young C, Young E, Aged C, Aged E, respectively) (Figure 3E).

#### 3.3.3. Pups’ Body Weight

Pups’ body weight (BW) for each group was as follows (1096.2 ± 15.7, 1055.7 ± 17.2, 1027.6 ± 11.1, 957.2 ± 11.5 mg, mean ± SEM, Young C, Young E, Aged C, Aged E, respectively). BW did not differ significantly between Young C and Young E, while BW was significantly lower in Aged E compared to Aged C (*p* < 0.0001), lower in Aged E compared to Young E (*p* < 0.0001), and lower in Aged C compared to Young C (*p* < 0.0001) (Figure 3F). As for the average body weight of pups to one dam (Av. BW) in each group was as follows (1097.4 ± 26.5, 1059.5 ± 27.7, 1044.8 ± 27.9, 953.8 ± 20.7 mg, mean ± SEM, Young C, Young E, Aged C, Aged E, respectively). Av. BW was not significantly different between Young C and Young E, but was significantly lower in Aged E than in Aged C (*p* < 0.05) (Figure 3E). In addition, Av. BW did not differ between Young C and Aged C, but Av. BW was significantly lower in Aged E than in Young E (*p* < 0.05), (Figure 3G).

### 3.4. Presence of Endometriosis Lesions in Both Endometriosis Groups

Endometriosis lesions were found in locations directly attached to or in close proximity to the uterus and other reproductive organs (Figure 4A). The total weight of endometriosis lesion per dam was significantly lower in Aged E compared to Young E (123.2 ± 19.6, 65.2 ± 9.8, mg, mean ± SEM, *p* < 0.05, Young E, Aged E, respectively) (Figure 4B); however, not a single mouse was found to have any lesions at all.

## 4. Discussion

The novel concept of this study was to examine the effects of “chronic exposure to endometriosis” prior to pregnancy on the perirenal outcome, in light of the recent escalation in the age at which women conceive [18,19,20]. In this study, we established for the first time a novel model to examine the impact of endometriosis on fertility and peritoneal outcome, taking into account the time from disease onset to pregnancy. Endometriosis was induced or sham-operated at 7 weeks of age, and the mice were mated for 6 weeks each, from 7 to 12 weeks of age in the Young study and from 13 to 18 weeks of age in the Aged study. The rate of resorption was higher in the Aged endometriosis group compared to the Aged control group, although such a difference was not observed in the Young study. Similarly, pups’ body weight was significantly lighter in the endometriosis group compared to the control in the Aged study, while there was no significant difference in the Young study. These results suggest that in addition to the presence of endometriosis per se, “chronic exposure to endometriosis” prior to pregnancy affects perinatal outcomes.

In this study, we found that the number of resorptions was significantly higher in the endometriosis group than in the controls only in the Aged but not in the Young study, suggesting that the occurrence of resorption is not related to the presence of endometriosis per se, but to “chronic exposure to endometriosis” prior to pregnancy. In humans, endometriosis is known to be associated with miscarriage [2,24], but reports in rodent models have been controversial: only one rat study [15] has so far reported that endometriosis increases resorption, while other mouse [14,16] and rat [13] studies have shown that the rate of resorption is not affected by endometriosis. Previously, in the advance maternal age mouse model, Hirata et al. [25] have shown that the number of resorptions increased with increasing maternal age, although their maternal age (11–13 M) is much older than that of our model (13–18 W). These findings suggest that the increased number of resorptions in our model may be due to a combination of two risk factors: advanced maternal age and “chronic exposure to endometriosis”. Whether this is due to an abnormality in the oocyte, such as anomalies in postovulatory oocyte structure and preimplantation embryo development, as reported in a study of endometriosis in rats [15], or problems in the uterus, requires further study.

The current study also succeeded in reproducing the SGA phenotype of endometriosis-associated pregnancy by extending the time between the induction of endometriosis and initiation of mating, or the duration of “chronic exposure to endometriosis” prior to pregnancy. We found that in the Young study, pups’ body weight was not significantly different between the controls and endometriosis groups while in the Aged study, pups’ body weight was significantly lighter in the endometriosis group than in the controls. It is well known that patients with endometriosis are at high risk of developing SGA [2,24,26]; however, none of the previously reported mouse models have reproduced this phenotype of SGA [12,13,14,15,16], and this was also true in our Young study. Advanced maternal age is known to be a risk factor for SGA in humans [27,28] and studies in mice have reported that SGA occurs at the age of 38 weeks [29] and 56 weeks [25]. In contrast, the present study showed that in mice with endometriosis, SGA appears at 13–18 weeks of age, an age at which SGA does not appear in control mice without endometriosis. These findings suggest that “chronic exposure to endometriosis” prior to pregnancy may predispose to SGA even with a slight increase in maternal age. This novel mouse model is therefore useful in elucidating the mechanism of SGA development in pregnancies complicated by endometriosis.

There are a number of epidemiological studies showing an increased risk for preterm labor in cases of endometriosis [2,30,31,32,33,34]: however, until now, no animal endometriosis model has successfully reproduced the occurrence of preterm labor. Our mouse model of endometriosis showed, for the first time, that endometriosis may potentially predispose to preterm delivery, and therefore this model is expected to be useful in future studies on the association between endometriosis and preterm labor. Unfortunately, however, because of its very low incidence, it is not concluded that the incidence of preterm labor is higher in the endometriosis with statistical significance. The effect of maternal age and duration of endometriosis prior to pregnancy on the risk of developing preterm labor was also not clear in the current study. Since advanced maternal age is also associated with preterm labor [27,28,35], it would be worthwhile to continue these studies that take into account the influence of maternal age and duration of endometriosis prior to pregnancy.

Clinically, the relation between endometriosis and infertility is well known, with numerous epidemiological demonstrating an increased risk of infertility in endometriosis cases [36,37,38]. It has also been demonstrated in rodent models that endometriosis reduces pregnancy rates [13,16]. Furthermore, several clinical data indicate that the decline in pregnancy rates is further exacerbated when older age is added to endometriosis [39,40]. In the present study, however, contrary to our expectations, the negative impact of endometriosis on pregnancy rates appeared significant in young mice and not in aged mice. This finding in young mice supports previous similar rodent models [13,16], and the mechanism could be that acute intra-abdominal inflammation trigged by the procedure of endometriosis induction may have affected ovulation or oocyte transport. On the other hand, as to why the pregnancy rate in the endometriosis group was not significantly lower in the Aged study, it could be that our “Aged” age setting is still much younger than many other similar models [41,42], so the age factor may not have manifested itself. In terms of the duration of “chronic exposure to endometriosis” prior to attempting to conceive, it is possible that the study period was not long enough for infertility to develop, and further reexamination with a longer duration may be warranted.

This mouse model may also be useful in optimizing health care methods during the long preconception period. Although there are several epidemiological studies on the effects of lifestyle factors such as exercise and weight restriction [43], and diet factors such as high-fat diets and phytoestrogens on endometriosis, few animal studies have been reported [8]. Supplementation with inositol and melatonin, for example, has been reported to contribute to the normalization of glucose tolerance during pregnancy and is expected to positively impact fetal outcomes [44,45]. In the future, our current model could be used to study the impact of lifestyle, diet, and supplementation on the chronic exposure to endometriosis.

Our study has several limitations. First, this study did not examine the mechanisms that cause resorption or fetal growth restriction. The authors are currently performing morphological comparisons and comprehensive genetic analyses using the implantation site and placenta, as the reviewer suggested, as well as maternal blood, urine, blood vessels, kidneys, uterine muscle, cervix, fetal brain, intestines, etc., which will be reported in the future. Second, since this study was conducted in rodents, the structure of the uterus and placenta is different from that of humans, and pathological conditions such as placenta previa cannot be examined. Finally, this study did not examine the prognosis of the dams in the postpartum period or the long-term prognosis of the pups because the dams were sacrificed prior to delivery in order to obtain fetal information. In the future, we would like to examine the effects of endometriosis in preconception period on the post-partum conditions of mothers and growth of infants.

## 5. Conclusions

In this study, we established a mouse model to examine the effect of chronic exposure to endometriosis on pregnancy outcome. Further modifications to the model may provide new insights, and the addition of molecular biological techniques may elucidate the mechanisms that cause adverse pregnancy outcome, and these findings can be applied to establish appropriate management methods for endometriosis in the preconception period.

## Figures and Tables

**Figure 1 biomedicines-10-02627-f001:**
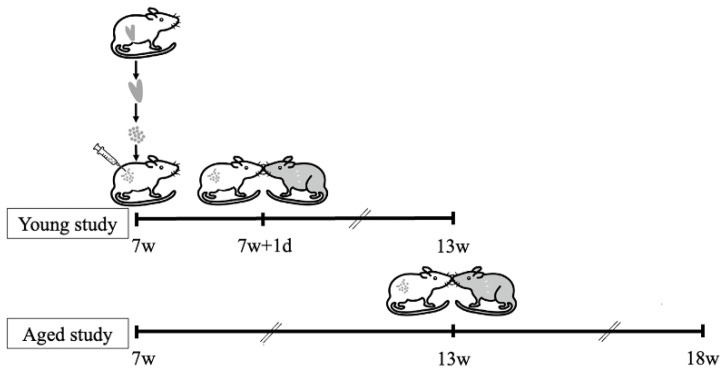
**A schematic diagram showing the flow of the two studies (Young study and Aged study).** This diagram shows experiment of the two studies (Young study and Aged study). Day 0 represents the day endometriosis was established. In the Young study, mating was initiated on day 1 and repeated every 4 days until day 42; in the Aged study, mating was initiated on day 43 and repeated every 4 days until day 84. To describe the experiments based on their week age, endometriosis was established or sham-operated at 7 weeks of age, and the mice were mated for 6 weeks each, from 7 to 12 weeks of age in the Young study and from 13 to 18 weeks of age in the Aged study.

**Figure 2 biomedicines-10-02627-f002:**
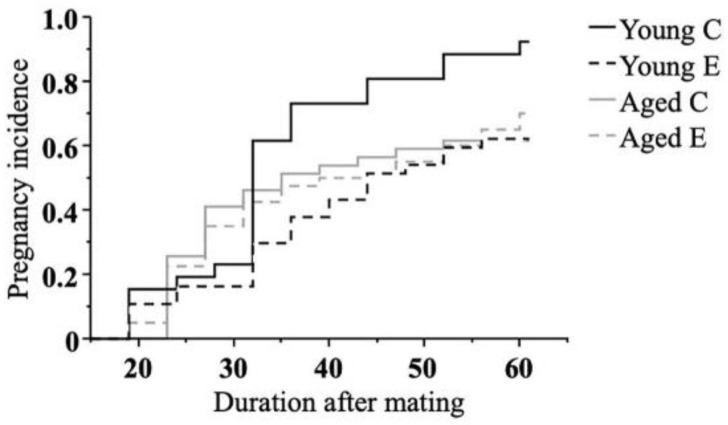
**Kaplan–Meier survival graph showing pregnancy incidence.** Mating was initiated on day 1 or day 43 and repeated every 4 days until day 42 or day 84 (Young study, Aged study, respectively). Mice that did not conceive by the last mating were considered infertile. Infertile mice were sacrificed at 900 am on day 60 or 102 (Young study, Aged study, respectively). Thus, the observation time was 60 days after the initiation of mating in both studies. The Young C group conceived significantly earlier (log-rank and generalized-Wilcoxon, *p* < 0.05) than the Young E group (Young C; n = 26, Young E; n = 34, Aged C; n = 40, Aged E; n = 39).

**Figure 3 biomedicines-10-02627-f003:**
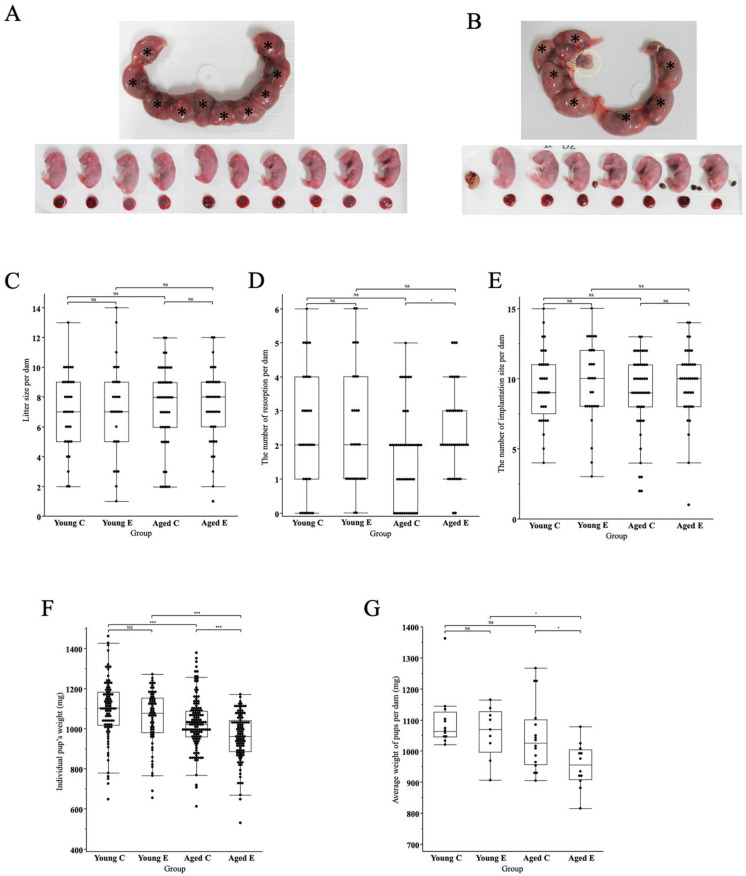
(**A**) **A representative macroscopic image of the uterus, pups and placenta of a control mouse.** The uterus was obtained at 18 dpc. Pups were cleaned from the uterus, pictured, and weighted. * indicates fetus in the uterus. (**B**) **A representative macroscopic image of the uterus, pups, resorptions and placenta of an endometriosis mouse.** The uterus was obtained at 18 dpc. Pups were cleaned from the uterus, pictured, and weighted. * indicates fetus in the uterus, yellow dotted circle is the lesion. (**C**) **A box-and-whisker graph of litter size.** Litter size was counted, and analyzed using the Wilcoxon test. Litter size was not significantly different between the groups. NS: *p* > 0.05. (**D**) **A box and whisker graph of resorption.** The number of resorptions was counted and analyzed using the Wilcoxon test. The number of resorptions was significantly higher in Aged E compared to Aged C while there was no difference between the other groups. NS: *p* > 0.05, *: *p* < 0.05. (**E**) **A box-and-whisker graph of implantation site.** The number of implantation sites was analyzed using the Wilcoxon test. Implantation sites were calculated as the sum of litter size and the number of resorptions. The number of implantation sites did not differ significantly between the groups. NS: *p* > 0.05 (Young C; n = 33, Young E; n = 29, Aged C; n = 47, Aged E; n = 39 for a, b, c). (**F**) **A box-and-whisker graph of pup body weight.** Body weights (BW) of pups were measured and analyzed using the Wilcoxon test. Pups with a litter size of ≤ 3 or ≥ 13 were excluded. BW did not differ significantly between Young C and Young E, while BW was significantly lower in Aged E compared to Aged C, lower in Aged E compared to Young E and lower in Aged C compared to Young C. NS: *p* > 0.05, ***: *p* < 0.0001 (Young C; n = 91, Young E; n = 65, Aged C; n = 133, Aged E; n = 103). (**G**) **A box-and-whisker graph of average body weight of pups to one dam.** The average body weight of pups (Av. BW) to one dam was calculated and compared using the Wilcoxon test. Dams with litter size ≤ 3 or ≥ 13 were excluded. Av. BW was not significantly different between Young C and Young E, but was significantly lower in Aged E compared to Aged C. In addition, Av. BW did not differ significantly between Young C and Aged C, but was significantly lower in Aged E compared to Aged E. NS: *p* > 0.05, *: *p* < 0.05 (Young C; n = 12, Young E; n = 9, Aged C; n = 16, Aged E; n = 12).

**Figure 4 biomedicines-10-02627-f004:**
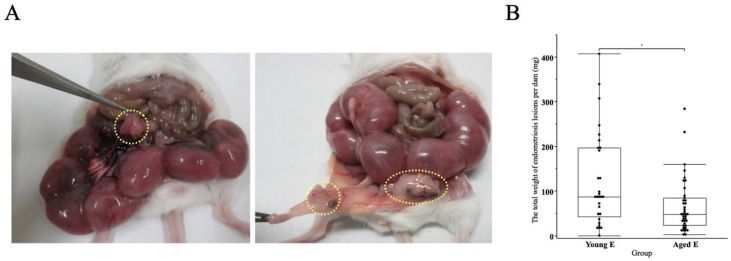
(**A**) **Representative macroscopic images of**
**anatomical relation of uterus and the endometriosis lesions**
**(yellow dotted circles****).** The sacrifice was performed at 18 dpc. (**B**) **A box-and-whisker graph of the total wight of endometriosis lesions per dam.** The endometriosis lesions were weighted and analyzed using the Wilcoxon test. The total weight of endometriosis lesions per dam was significantly lower in Aged E group compared to Young E group (*p* < 0.05). *: *p* < 0.05 (Young E; n = 29, Aged E; n = 39).

**Table 1 biomedicines-10-02627-t001:** Preterm labor.

	Young C	Young E	Aged C	Aged E
**Total # of dam**	33	29	47	39
**# of dam delivered before dpc18**	0	2	0	1
**% of dam delivered before dpc18**	0	6.9	0	2.6

## Data Availability

Not applicable.

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
