# Peer review of "Impact of Chronic Exposure to Endometriosis on Perinatal Outcomes: Establishment of a Mouse Model"

_biomedicines, 2022, doi:10.3390/biomedicines10102627_

Round 1
Reviewer 1 Report
The authors established novel mouse model to examine the effect of chronic exposure to endometriosis on pregnancy outcome. They demonstrated that the number of resorptions was significantly higher, and body weight was significantly lower, in aged studies group than that in control. They claimed that chronic exposure endometriosis prior to pregnancy affect perinatal outcome. I believe that this study is interesting and provide attractive informations to readers. However, this study has some concerns. This study needs some minor revisions to accept for this journal.
I strongly recommend that the authors investigate morphological changes of implantation site and placenta to clarify the mechanisms of these findings.
Author Response
List of Point-by-Point Response to the Reviewer’s Comments
Response to the comments of reviewer #1
Comment: I believe that this study is interesting and provide attractive information to readers. However, this study has some concerns. This study needs some minor revisions to accept for this journal.
Response: The authors appreciate the favorable comment.
Comment: I strongly recommend that the authors investigate morphological changes of implantation site and placenta to clarify the mechanisms of these findings.
Response: The authors appreciate the comment. The limitation of the current manuscript is that it is only an observational study of phenotypes and does not perform any functional analysis. The authors are, however, currently performing morphological comparisons and comprehensive genetic analyses using the implantation site and placenta as the reviewer suggested, as well as maternal blood, urine, blood vessels, kidneys, uterine muscle, cervix, fetal brain, intestines, etc., which will be reported together as a separate independent paper. The authors have added the following descriptions in the revised manuscript (see L266-270).
(L266-270): The authors are currently performing morphological comparisons and comprehensive genetic analyses using the implantation site and placenta, as the reviewer suggested, as well as maternal blood, urine, blood vessels, kidneys, uterine muscle, cervix, fetal brain, intestines, etc., which will be reported in the future.

Reviewer 2 Report
I read with great interest the manuscript, which falls within the aim of this Journal. In my honest opinion, the topic is interesting enough to attract the readers’ attention. Nevertheless, authors should clarify some points and improve the discussion, as suggested below. Authors should consider the following recommendations:
In my opinion you have to improve the paper referign to the lifestyle impact in pts with endometriosis.
I also suggest you to refer how has been demonstrated the multiple inositols role in infertile copule also with endometriosis pts
I suggest you to add a small paragraf about the multiple inositols rolefrom induction of ovulation to menopausal transition, to prevent gestational diabetes.
I suggest you read and cite these articles:
Impact of lifestyle and diet on endometriosis: a fresh look to a busy corner.
Translational animal models for endometriosis research: a long and windy road.
Myo-inositol and melatonin in the menopausal transition.
Myoinositol plus α-lactalbumin supplementation, insulin resistance and birth outcomes in women with gestational diabetes mellitus: a randomized, controlled study.
Author Response
List of Point-by-Point Response to the Reviewer’s Comments
Response to the comments of reviewer #2:
Comment: I read with great interest the manuscript, which falls within the aim of this Journal. In my honest opinion, the topic is interesting enough to attract the readers’ attention.
Response: The authors appreciate the favorable comment.
Comment: In my opinion you have to improve the paper referign to the lifestyle impact in pts with endometriosis. I also suggest you to refer how has been demonstrated the multiple inositols role in infertile copule also with endometriosis pts. I suggest you to add a small paragraf about the multiple inositols rolefrom induction of ovulation to menopausal transition, to prevent gestational diabetes.
I suggest you read and cite these articles:
Impact of lifestyle and diet on endometriosis: a fresh look to a busy corner.
Translational animal models for endometriosis research: a long and windy road.
Myo-inositol and melatonin in the menopausal transition.
Myoinositol plus α-lactalbumin supplementation, insulin resistance and birth outcomes in women with gestational diabetes mellitus: a randomized, controlled study.
Response: The authors appreciate the comment. The following paragraph has been added in the revised manuscript and the authors have cited the articles as suggested (See L256 - 264 and references 8, 43, 44 and 45).
(L256-264): This mouse model may also be useful in optimizing health care methods during the long preconception period. Although there are several epidemiological studies on the effects of lifestyle factors such as exercise and weight restriction (8), and diet factors such as high-fat diets and phytoestrogens on endometriosis, few animal studies have been reported (43). Supplementation with inositol and melatonin, for example, has been reported to contribute to the normalization of glucose tolerance during pregnancy and is expected to positively impact fetal outcomes (44,45). In the future, our current model could be used to study the impact of lifestyle, diet, and supplementation on the chronic exposure to endometriosis.
